# Adverse Effect of Polystyrene Nanoplastics in Impairing Glucose Metabolism in Liver Injury

**DOI:** 10.3390/ijms26104870

**Published:** 2025-05-19

**Authors:** Raden Joko Kuncoroningrat Susilo, Manikya Pramudya, Firli Rahmah Primula Dewi, Windy Seftiarini, Dewi Hidayati, Aunurohim Aunurohim, Vuanghao Lim, Mochammad Aqilah Herdiansyah, Alfiah Hayati

**Affiliations:** 1Nanotechnology Engineering, Faculty of Advanced Technology and Multidiscipline, Universitas Airlangga, Surabaya 60115, Indonesia; joko.kuncoroningrat@ftmm.unair.ac.id; 2Department of Biology, Faculty of Science and Technology, Universitas Airlangga, Surabaya 60115, Indonesia; manikya.pramudya@fst.unair.ac.id (M.P.); firli.rahmah@fst.unair.ac.id (F.R.P.D.); moch.aqilah.herdiansyah-2024@fst.unair.ac.id (M.A.H.); 3Graduate School of Biotechnology, Universitas Gadjah Mada, Yogyakarta 55281, Indonesia; windyseftiarini@mail.ugm.ac.id; 4Department of Biology, Faculty of Science and Data Analytics, Institut Teknologi Sepuluh Nopember, Surabaya 60111, Indonesia; d_hidayati@bio.its.ac.id (D.H.); aunurohim@bio.its.ac.id (A.A.); 5Advanced Medical and Dental Institute, Universiti Sains Malaysia, Bertam, Kepala Batas 13200, Penang, Malaysia; vlim@usm.my

**Keywords:** nanoplastics, apoptosis, glucose metabolism, healthy life, liver

## Abstract

Polystyrene nanoplastics (PS-NPs) are result from the degradation of plastic and have diameters ranging from 1 nm to 100 nm. The objective of this study is to provide information on the adverse effects of PS-NPs with in vitro and in vivo analyses of liver injury. An in vitro study was conducted using confocal microscopy, flow cytometry, and MTT test analysis. An in vivo study was conducted to determine apoptosis levels, glucose metabolism gene expressions, liver enzymes, and liver histology. Data were analyzed using GraphPad Prism software 10.2.1. The in vitro study showed the absorption of PS-NPs in the cell cytoplasm, the percentage of apoptosis, 3t3, and the WiDr cell lines’ viability. The in vivo analysis showed that PS-NPs can stimulate liver injuries, such as inducing the elevation of liver enzymes, necrosis, edema, inflammation, and the dilatation of the portal vein diameter. High levels of caspase-3, caspase-9, and Bax were detected, as well as the expression of several genes including PI3K, AKT, PEPCK, GLUT2, and PK. In conclusion, the in vitro analysis showed the detrimental effects of PS-NPs on cells, such as high levels of apoptosis and low cell viability, while the in vivo studies displayed the impairment of liver tissue and disturbances in glucose metabolism regulation.

## 1. Introduction

Plastics have become essential to many human activities, such as shopping and working, as well as in household materials. Many plastics are not recycled wisely and have an adverse effect on the environment. About 99 million tons of plastics are produced from human activities [1]. Plastics are divided based on their main materials, such as polyethylene (PE), polypropylene (PP), polystyrene (PS), and polyvinyl chloride (PVC) [2]. Among these materials, PS is the one most commonly consumed by people, as the packaging of many daily consumable products comes from this material [3]. Polystyrene nanoplastic (PS-NP) particles are produced from the degradation of plastics by abrasion, ultraviolet light, and biodegradation [4]. PS-NPs particles have a size range between 1 and 100 nm and are widely detected in the food chain and in the environment, e.g., in water, sediment, air pollution, and drinking water. PS-NPs are regarded as a key problem in people’s lives as they are prevalent in the lived environment. Several studies have shown the hazardous effects of PS-NPs, such as oxidative stress, mitochondrial dysfunction, DNA damage, inflammation, and autophagy disruption. PS-NPs have the ability to accumulate in organisms, especially in aquatic organisms. PS-NPs can also persist in aquatic and terrestrial ecosystems due to their small size and stability. Moreover, information regarding the toxic effects of microplastics in organisms suggests that they are damaging to health, causing gastrointestinal problems, immune system disruption, and respiratory issues; they are also potentially carcinogenic. Several studies have shown that microplastics can break down into smaller particles until they reach nano form via certain mechanisms, including mechanical abrasion, photooxidation, biodegradation, chemical degradation, and cryo-milling. This condition also has nanoplastic effects, especially regarding PS-NPs, which are derived from microplastics. Several effects of PS-NPs need to be explored in the research, as this substance represents a major problem in health in relation to microplastics.

PS-NPs can easily enter the body, causing toxic effects on health through ingestion while via contaminated food, inhalation, as well as potential dermal contact [5,6]. They can directly interact with cell membranes depending on the surface properties and size of particles [7]. PS-NPs particles can enter cells through the endocytosis pathway [8,9,10,11]. Molecular transport through endocytosis occurs through an adhesive interaction pathway between PS-NPs and the cell membrane, which is mediated by cell receptors. The endocytosis mechanisms, such as clathrin and caveolae, act as mediators for phagocytosis and macropinocytosis toward PS-NPs [12,13]. The type of cell also influences PS-NP absorbance into the cells. PS-NPs can also penetrate alveolar epithelial cells through a non-endocytic pathway such as cell-penetrating peptides (CPPs) and pH (low) insertion peptides (pHLIPs) [14].

Plastic polymers of ≥700 nm have been detected at an average of 1.6 µg/mL in human blood. The diameter of blood capillaries is 5–8 µm, which allows particles with <1 µm in size to circulate in microvascular areas, affecting fluid dynamics [15]. PS-NPs have been detected not only in the bloodstream but also in the liver, pancreas, heart, gonads, and brain [16,17,18]. The presence of PS-NPs in biological tissues has been associated with side effects, such as increased oxidative stress, inflammation, cell damage, and apoptosis [5,19]. PS-NPs is metabolized by cytochrome P450 (CYP450) in several organs, such as the liver, intestine, kidneys, and lungs, making it less toxic. However, the side effects of this action still create reactive oxygen species (ROS), which directly cause injury to nuclear DNA and mitochondria [20,21,22,23]. PS-NPs can easily stimulate TNF receptor-associated death domain (TRADD) and Fas-associated protein with death domain (FADD) to activate the extrinsic apoptosis pathway via caspase-8 activity [24,25]. The injury can also increase the apoptosis percentage in the cell through the intrinsic apoptosis pathway via activation of caspase-9 with caspase-3 execution. Moreover, exposure to PS-NPs leads to impaired glucose metabolism, with results including high lactic acid levels, low insulin levels, and low pyruvate kinase (PK) secretion [26,27].

The liver plays a central role in the metabolism of xenobiotic agents such as environmental pollutants, hydrocarbons, food additives, pesticides, and drugs [28]. All of these agents have the potential to undermine liver cells via oxidative stress [29,30]. The liver also provides glucose storage for excessive glucose in the bloodstream, which is taken up into the cell as glycogen [31]. PS-NPs can stimulate various signaling proteins to mediate insulin pathways through the excessive induction of insulin receptor substrate-1 (IRS-1) phosphorylation in liver cells. This mechanism leads to high glucose uptake from the bloodstream beyond the normal activity state [32]. This condition can disrupt glucose metabolism in the liver and tends to convert surplus glucose into fat [33,34]. This cascade reduces the contribution of glycogenolysis to glucose production in the body. Moreover, the liver also plays a role in gluconeogenesis while in a fasting state, when glucose is required for energy. This mechanism maintains glucose levels in the blood and is important for glucose homeostasis. Recent studies have shown that impaired glucose homeostasis tends to create liver toxicity, which can ultimately result in severe injury [6,32,33,35,36,37,38].

The adverse effects of PS-NPs pose significant risks to the body that require further exploration via in vivo and in vitro experiments. These effects are important in preventing risk from plastic usage, especially in terms of liver toxicity. Thus far, limited information is available regarding the molecular mechanisms of PS-NPs in inducing signaling pathways related to apoptosis and glucose metabolism, ultimately leading to liver toxicity. This study aims to investigate the hazardous effect of PS-NPs exposure in cells on the apoptosis pathway and the impairment of liver glucose metabolism via molecular pathways, including mRNA analysis, until liver injury.

## 2. Results

### 2.1. Effect of PS-NPs on Cell Lines

As shown in Figure 1A,B, the combination of Nile Red and DAPI staining under a confocal microscope indicated that PS-NPs can enter the nucleus from the 3t3 and WiDr cells. Specifically, induction with 100 µL/mL of PS-NPs resulted in strong expression inside the cell. Nile red staining was used to display the PS-NPs, and then DAPI was used for 3t3 and WiDr cell staining. The 10 µL/mL group showed early endocytosis by the 3t3 and WiDr cells against PS-NPs. However, the PS-NPs became endocytic in the cell membrane in the 25 and 50 µL/mL groups and had already entered the cytoplasm of 3t3 and WiDr cell lines in the 100 µL/mL group. This result emphasizes the endocytosis process in the 3t3 and WiDr cells, which is proven by the presence of many PS-NPs in the cell cytoplasm. As shown in Figure 1C, the MTT assay displayed that the number of living cells decreased in a dose-dependent manner, with 100 µL/mL having the lowest living cell percentage of 28.92%. The relatively high standard deviation shown in the figure was due to the elevation of cellular stress and the potential aggregation of PS-NPs, which resulted in uneven cell exposure across the full area. PS-NPs could effectively kill the 3t3 cell line population. Meanwhile, Figure 1D shows that the cell viability of the WiDr cell line declined in a dose-dependent manner. The lowest percentage of cell viability was at a dose of 100 µL/mL, with only 24.24%. These results indicate that PS-NPs are potentially toxic against cell lines of both normal cells and cancer cells.

### 2.2. Effect of PS-NPs on Flow Cytometry Assay

Treatment with PS-NPs increased the percentage of early and late apoptotic cells, according to the flow cytometry tests. This result was observed via Annexin V and PI staining. As shown in Figure 2A, the induction of PS-NPs could increase early apoptosis and late apoptosis from 3t3 cells. The strongest expression of early apoptosis was shown at a dose of 100 µL/mL, but the highest expression for late apoptosis was displayed at a dose of 50 µL/mL. It seems that the induction of PS-NPs had toxic effects on the 3t3 cell line. Figure 2B indicates that PS-NPs were toxic, having the ability to elevate the percentage of apoptotic cells. A dose of 10 µL/mL showed higher expression levels of early apoptosis and late apoptosis markers compared to 100 µL/mL, which seemed to predominantly induce necrosis. Figure 2C showed that a dose of 100 µL/mL had the highest percentage of total apoptosis, which is the total combination of early and late apoptosis cells after PS-NPs induction, with a value of 46.17%. Meanwhile, Figure 2D had the highest total apoptosis percentage from a dose of 10 µL/mL with 5.93%. It seems that, in WiDr cells, PS-NPs exposure has only slight toxic effects compared to the 3t3 cell line. These results mean that both 3t3 cells and WiDr could undergo a toxic effect from PS-NPs induction.

### 2.3. Effect of PS-NPs on Apoptosis Parameters

Several results of the ELISA test showed that PS-NPs significantly increased the levels of pro-apoptotic enzymes, such as caspase-3 (Figure 3A), caspase-9 (Figure 3B), and Bax (Figure 3C), compared with those in the control group. The 10 µL/kg group had a higher level of pro-apoptotic enzymes than the other treatment groups, with 1.34 ± 0.15 ng/mL caspase-3, 503.71 ± 24 ng/L Bax, and 151.28 ± 7.4 ng/L caspase-9. In the Fas gene, the 1, 5, and 10 µL/kg groups had increasing values compared to the control group. However, the treatment groups were not significantly elevated compared to the control group (Figure 3D).

### 2.4. Effect of PS-NPs on the mRNA Expression of Glucose Metabolism Genes

Several genes related to lipid metabolism were evaluated. PI3K expression (Figure 4A) in the 1, 5, and 10 µL/kg PS-NPs groups was significantly increased compared with the control group. The Akt gene (Figure 4B) was significantly increased in the 5 and 10 µL/kg PS-NPs groups in comparison with the control group but did not significantly increase in the 1 µL/kg group. PS-NP treatments significantly elevated PEPCK (Figure 4C) gene expression in the 5 and 10 µL/kg PS-NP groups compared with the control group. However, the 1 µL/kg PS-NPs group did not show a significant increase in expression. PS-NPs also affected GLUT2 expression (Figure 4D), which showed significantly elevated levels in the 5 and 10 µL/kg groups but a significant decrease in the 1 µL/kg group. Furthermore, the PS-NPs treatments also significantly decreased the level of the PK gene (Figure 4E) in the 5 and 10 µL/kg PS-NPs groups; the 1 µL/kg PS-NPs group did not exhibit decreased expression.

### 2.5. Effect of PS-NPs on Liver Tissue

The histology analysis indicated that the PS-NPs treatment could cause several liver injuries, such as necrosis, edema, and the inflammation of cells (Figure 5A). The 1 µL/kg group showed edema and necrotic cells. However, the 5 µL/kg group showed necrotic cells only. High doses of PS-NPs in the 10 µL/kg group impaired the liver, as evidenced by necrotic, edematous, and inflammatory cells. Figure 5B shows that normal cells displayed a significant reduction compared with the control group. The number of edema cells (Figure 5C) significantly increased, with the highest value in the 10 µL/kg group (5.42 ± 0.49 cells). The percentage of necrotic (Figure 5D) and Kupffer cells (Figure 5E), as well as vena porta (Figure 5F), significantly increased in a dose-dependent manner in comparison with the control group. Furthermore, vena centralis (Figure 5G) slightly increased compared with that in the control group. As shown in Figure 5H,I, PS-NPs also affected ALT and AST levels. The results show that the induction of PS-NPs can significantly increase ALT and AST levels in comparison with the control group. The highest ALT level was 22.1 µg/mL, which occurred in the P1 group. AST levels also reach a maximum (44.3 µg/mL) in the P1 group.

## 3. Discussion

PS-NPs are hazardous pollutants in water and on land. This study explores the effect of PS-NPs on the apoptosis pathway, a pathway that ultimately results in liver injury. This study uses commercial PS-NPs, which are 100 nm in size and monodisperse. This material is often used in nanotoxicology studies. In comparison with environmental nanoplastics, commercial PS-NPs have characteristics such as a size and surface charge area that are representative of environmental ones, especially those resulting from the breakdown of plastic food containers, disposable cutlery, industrial products, and packaging. Environmental nanoplastics have a size range of 100–200 nm and also exhibit a negative surface charge area due to weather and oxidation processes in the environment. Thus, commercial PS-NPs are relevant to the model, reflecting plausible biological responses in in vitro and in vivo studies [39,40]. The viability of 3t3 and WiDr cells was reduced after PS-NPs treatment, which was caused by the toxicity of PS-NPs. We used 3t3 and WiDr cells to determine the general cytotoxicity and apoptotic effects of PS-NPs on normal and cancer cells. Both of them exhibit rapid cell division and different cellular responses, especially for cancer cells that are more sensitive to nanoparticle-induced stress. Some studies show that PS-NPs can enter and accumulate easily in cell lines such as HKC, HL-7702, and A549 cells. They were also displayed in the confocal microscope analysis using DAPI and Nile red [41]. Toxicity can increase ROS in the cells and disrupt the cell structure until apoptosis and necrosis [42]. The combination of Nile red and DAPI staining indicated that PS-NPs entered the cell via endocytosis. The process started at the cell membrane, and the cells caught the PS-NPs to deliver them into the cytoplasm and nucleus. The MTT assay was conducted to determine the proper dose to eliminate 3t3 and WiDr cells. The decreased viability of 3t3 and WiDr cells indicated that PS-NPs has cytotoxic potential [43]. Dose-dependent changes caused by the PS-NPs treatment indicate that its toxicity potential increases with elevating doses.

Apoptosis is a mechanism of regulated cell death that is initiated by internal stress and toxic substances. The flow cytometry analysis conducted in this study shows that apoptosis occurred at many of the doses used, although the highest dose of PS-NPs produced a high percentage of necrosis [44]. High doses of PS-NPs could lead to the apoptosis of 3t3 and WiDr cells because they affect the intrinsic pathway in the apoptosis cascade. Stress in the mitochondria can be induced by radiation, chemotherapy drugs, and toxic agents, such as PS-NPs [45]. This substance leads to elevated ROS from the mitochondria, thereby triggering the activation of pro-apoptosis enzymes such as Bax, caspase-9, and caspase-3 [46]. Additionally, ROS can activate p53 and c-Jun N-terminal kinase (JNK), which activates Bax proteins to inhibit antiapoptotic protein action. Several pro-apoptosis enzymes displayed increasing levels after treatment with PS-NPs. The Bax levels increased because of the activation of apoptosis activity [23].

The mitochondrial membrane undergoes depolarization and the opening of Bax channels, allowing for the release of cytochrome-c (cyt-c) into the cytosol [47]. Subsequently, cyt-c initiates the apoptosome complex in the cytosol along with apoptotic protease activating factor-1 (Apaf-1) and procaspase-9, as formed before activated caspase-9. The activation of caspase-9 initiates the intrinsic apoptosis pathway and becomes elevated when 3t3 cells are induced by NP [48]. The activation of caspase-9 leads to the formation of caspase-3, which acts as executor [49]. The apoptosis pathway was also influenced by the Fas cell surface death receptor. The activation of the Fas receptor, which binds to the Fas ligand (FasL), could lead to an extrinsic apoptosis pathway [27,50]. The pathway also inhibits Bcl-2 activation, thereby increasing Bax levels and triggering intrinsic apoptosis pathways [51]. Intrinsic and extrinsic apoptosis pathways rely on caspase-3 to execute apoptosis until the cell nucleus shrinks and disappears [47].

PS-NPs also had significant effects on glucose metabolic genes, such as PI3K, AKT, GLUT2, PEPCK, and PK. The PI3K gene was activated when insulin bound to the insulin receptor (IR) [52]. Furthermore, *PI3K* stimulates the activation of AKT, which leads to the inactivation of the forkhead box O1 (FoxO1) transcription factor [53]. GLUT2 plays a role in glucose mobilization into the cell. Increasing levels of PI3K, AKT, and GLUT2 are signs of high activity as glucose is taken into the cell. The upregulation of PI3K and AKT expression disturbs an insulin signaling pathway, which stimulates the translocation of GLUT2 from the cell cytoplasm into the cell membrane. This condition can disturb glucose homeostasis, e.g., through excessive glucose uptake, glycogen storage, and lipogenesis [54]. The PEPCK signal acts as the primary gluconeogenesis enzyme, which initiates gluconeogenesis from non-carbohydrates in the liver. This signal converts oxaloacetate into phosphoenolpyruvate (PEP) [55]. In the liver, FoxO1 can collaborate with the co-activator gamma receptor-activated proliferator peroxisome 1α (PGC-1α) to enhance the expression of genes encoding gluconeogenesis, including PEPCK and glucose-6-phosphatase (G6Pase) [34]. Thus, increased hepatic glucose production occurs through the increased regulation of PEPCK and G6Pase gene expression [56]. G6Pase activation enhances the conversion of glucose-6-phosphate (G6P) into glucose, thereby increasing glucose availability in the body [57]. Otherwise, *PK* is involved in the conversion of PEP into pyruvate during glycolysis [58]. The elevation of *PEPCK* after PS-NPs treatment stimulates gluconeogenesis in a high state to synthesize glucose and reduces glycolytic flux [59]. This gluconeogenesis can reduce the glycolysis pathway, which is displayed by the low expression of *PK*. This imbalanced state leads to the accumulation of glucose and lipids within the liver, which contributes to liver steatosis, oxidative stress, and liver injury [60]. This study explores the relationship between apoptosis and glucose metabolism after PS-NP exposure. Excessive ROS from PS-NP metabolism can disturb the mitochondria until damage is caused, which activates intrinsic apoptosis. ROS can also alter the insulin pathway, leading to the disturbed activation of the PI3K/AKT pathway. In addition, glucose metabolism dysfunction stimulates the liver cells to uptake glucose and overstimulates glycogen storage and lipogenesis. This study reveals that PS-NPs induce the elevation of PEPCK expression, in contrast with the lowering of PK expression. This metabolic disorder causes an energetically imbalanced state in the cell. This condition constrains the cell, causing it to become stressed and run a cell death program such as apoptosis. This process can continuously disrupt metabolic function, which accelerates liver injury [27].

Elevated liver parameters exhibited the degree of injury to the liver. Higher values of liver parameters such as ALT and AST showed that the liver elevated metabolic activity to eliminate free radical factors from the liver tissue [61,62]. In this study, several ROS came from the failure of glucose metabolism, which was previously determined using qPCR analysis. This failure could increase the levels of excess glucose, which was stored in the liver tissue. Under these circumstances, the liver always tries to be homeostatic in a liver environment that needs more energy from the mitochondrial supply. The cascade of metabolic activity escalated ROS as a side product of the energy metabolism, which also plays an incrementally destructive role in liver cells [22,23]. The long-term effects of ROS exposure tend to cause irreversible liver injuries, such as fibrosis and cirrhosis. This glucose metabolism disorder affects cell conditions and leads to liver injury, such as edema and necrosis of cells [30,56]. Liver injury was also displayed by several inflamed cells, especially around the vena porta [61,63]. PS-NPs treatment could impair glucose metabolism, leading to a hyperglycemic state. This condition increases ROS production from the mitochondria and disrupts the cell structure [32].

## 4. Materials and Methods

### 4.1. Cell Culture (In Vitro Experiment)

Mouse embryonic fibroblast (3t3) and WiDr cell lines were obtained from the Parasitology Laboratory, Faculty of Medicine, Public Health and Nursing, Universitas Gadjah Mada, Yogyakarta, Indonesia. The 3t3 and WiDr cells were plated on a dish culture in Dulbecco’s Modified Eagle Medium (DMEM) (Gibco, USA) supplemented with an additional 10% fetal bovine serum (FBS, Gibco, Thermo Fisher Scientific, Waltham, MA, USA), 1% penicillin–streptomycin solution, and 0.5% fungizone; they were cultured at 37 °C in 5% CO_2_. The 3t3 and WiDr cells were treated with PS-NPs in dosages of 10 µL/mL, 50 µL/mL, and 100 µL/mL. After that, the 3t3 and WiDr cells were incubated for 48 h. The control group comprised 3t3 and WiDr cell lines without PS-NPs treatment.

### 4.2. Preparation of Confocal Microscope

The PS-NPs were purchased from Sigma-Aldrich (St. Louis, MO, USA; product number: 43302, carboxylated, ≥99% purity, dispersion in water) with a particle size of 100 nm for a standard deviation ≤ 0.01 μm. The PS-NPs also had a negative charge area of −50.80 ± 2.19 when dissolved in distilled water [39]. Slide preparation for the confocal microscope involved several steps. The cell culture was fixed with 4% formaldehyde for 20 min at room temperature. About 5 µL of the cell suspension was dropped gently onto a glass slide and smeared evenly. The object glass was prepared under a confocal microscope. Nile red and DAPI staining were added to the slide, which was then observed under a confocal microscope. Images were captured with an Olympus FV3000 confocal laser scanning microscope (Olympus Corporation, Tokyo, Japan). The control group comprised cells without PS-NPs induction, and the treatment groups were induced by PS-NPs with dosages of 10 µL/mL, 50 µL/mL, and 100 µL/mL.

### 4.3. Cell Viability Assay

The cytotoxicity of the PS-NPs was measured using a cell viability assay. It was estimated by measuring the mitochondrial reductase activity with an MTT reagent (≥98% purity, Sigma-Aldrich, Cat. No. M5655) with a concentration of 0.5 mg/mL in DMEM. The cells were seeded into 96-well culture plates at a density of 10,000 cells/well for all cell lines and incubated in a CO_2_ incubator. The control group was treated with no PS-NPs, and the treatment groups were specifically treated with PS-NPs 100 nm in diameter at 10 µL/mL, 50 µL/mL, and 100 µL/mL for 48 h. The cells were added along with 50 µL of the MTT substrate and incubated in a CO_2_ incubator in the dark for 48 h. The medium was removed, and the formazan crystals formed by the cells were dissolved using 100 µL of DMSO and transferred into a 96-well plate. Absorbance was read at 450 nm using a multi-mode microplate reader based on the kit protocol (SpectraMax iD3, Molecular Devices, Shanghai, China).

### 4.4. Flow Cytometry Assay

Here, 3t3 cells were collected, washed twice with PBS, and resuspended in annexin-binding buffer (0.1% sodium citrate, Triton X-100, and 5 μg/mL PI). Each cell sample was added to an annexin-V-FITC and propidium iodide (PI) buffer simultaneously to the cell suspension and stained for 10 min in the dark before the flow cytometry analysis. The control group was prepared with no PS-NPs, and there were four treatment groups, including the 10 µL/mL PS-NPs group, the 50 µL/mL PS-NPs group, and the 100 µL/mL PS-NPs group. Flow cytometry was performed using the BD FACScan flow cytometer automated system (Becton-Dickinson, Sunnyvale, CA, USA). Data related to the percentage of cells undergoing apoptosis and necrosis were analyzed using CellQuest Pro software version 6.0 and the ModFit LT 2.0 program.

### 4.5. Animals (In Vivo Experiment)

PS-NPs were induced using adult rats (*Rattus norvegicus* L.) of the Wistar strain. Twenty-eight rats aged 60 days and weighing 200 ± 15 g were obtained from the Animal Laboratory, Faculty of Pharmacy, Universitas Airlangga. The rats were individually housed in wire cages in the Animal Laboratory, Faculty of Science and Technology, Universitas Airlangga, Indonesia. The animal room was maintained at a controlled temperature of 25 °C ± 2 °C, relative humidity at 50% ± 5%, and light exposure cycle (12 h light–dark cycle). During testing, the rats were maintained at 80–90% of their free-feeding weight and drank water ad libitum. All procedures and treatments were approved by the Research Ethics Committee from the Faculty of Veterinary Medicine, Universitas Airlangga, Surabaya, Indonesia (1157/HRECC.FODM/X/2023).

### 4.6. Experimental Design

The experimental protocol involves treating animals with PS-NPs through oral administration using a 0.5 mL syringe. The treatment groups are divided as follows: the control group received distilled water (without PS-NPs) and the three treatment groups received PS-NPs at concentrations of 1 µL/kg, 5 µL/kg, and 10 µL/kg. After a 30-day treatment period, the rats were euthanized via an injection of 2% xylazine at a dose of 2 mg/kg BW and 10% ketamine HCl at a dose of 15 mg/kg BW intramuscularly, before being sacrificed. Hereafter, blood serum was collected via intracardiac and kept in a yellow-top Vacutainer Tube (OneMed, PT. Jayamas Medica Industri Tbk, Surabaya, Indonesia) for an enzyme-linked immunosorbent assay (ELISA). Liver samples were collected after euthanizing the rats and were kept in a 60 mL sample container (OneMed, PT. Jayamas Medica Industri Tbk, Indonesia) for real-time PCR and histological examinations.

### 4.7. ELISA Test

ELISA was utilized to detect and quantify specific proteins or antibodies in the serum to provide insights into the presence of specific biomarkers. The analysis of caspase-9, caspase-3, and Bax levels was performed using a commercial ELISA kit (Bioassay Technology, Shanghai, China). A total of 40 μL of blood serum was placed on the well plate. About 10 and 50 μL of antibodies were added to streptavidin–HRP. After 60 min of incubation at 37 °C, the plates were washed five times with a washing buffer. About 50 μL of substrate solutions A and B were added to every well. The plate was then incubated for 10 min at 37 °C in the dark. A total of 50 μL of stop solution was added to each well. Optical density (OD) values were recorded using a microplate reader (SpectraMax iD3, Molecular Devices, Shanghai, China) at a wavelength of 450 nm within 10 min.

### 4.8. Real-Time PCR Analysis

Real-time polymerase chain reaction (qPCR) was employed to measure gene expression levels and provide data on how PS-NPs treatment affects gene activity in the liver samples. qPCR was performed using a LightCycler 480 Instrument II (Roche Diagnostics GmbH, Mannheim, Germany). The expression levels of Fas, phosphatidylinositol-3 kinase (PI3K), protein kinase B (AKT), phosphoenolpyruvate carboxykinase (PEPCK), glucose transporter 2 (GLUT2), and PK genes were analyzed using qPCR. RNA was reverse transcribed using RT-PCR kits with an oligo d (T) 16 primer under standard conditions. Real-time PCR amplification was performed using a Light Cycler 480 with 2 mL of purified cDNA product, 0.5 mL of sense primer (10 pmol/mL), 0.5 mL of antisense primer (10 pmol/mL), 1 mL of Light Cycler FastStart DNA Master SYBR Green I, and 0.8 mL of MgCl_2_. Commercial glyceraldehyde phosphate dehydrogenase (GAPDH) primer sets were used for PCR amplification under the conditions recommended by the manufacturer. A list of primers from all of the genes is shown in Table 1.

### 4.9. Histology Analysis

The histological analysis involved examining liver tissue samples under a microscope to identify any structural changes or signs of toxicity; it allows for a detailed assessment of the impact of PS-NPs on the liver. The liver tissue was taken from the rats and immediately weighed. The histological analysis was performed by fixing the liver in a formalin buffer of 10% for 48 h. Tissue sections were placed in cassettes and washed thoroughly three times with K_3_PO_4_ solution (30 min for each wash) and twice with xylene (30 min for each). This tissue was also embedded in three times paraffin (1 h each). The tissue was cut to a 5 µm thickness. The tissue sections were stained with hematoxylin and eosin (H&E) and observed using a light microscope (Leica, Wetzlar, Germany). The observation results were evaluated using ImageJ version 1.53t (National Institutes of Health, Bethesda, MD, USA).

### 4.10. Statistical Analysis

Data related to the effect of the PS-NPs treatments are presented as the mean ± SD using one-way ANOVA. Statistical significance was calculated according to Tukey’s post hoc mean comparison test at a significance level of *p* < 0.05 using GraphPad Prism software.

## 5. Conclusions

This study demonstrates that PS-NPs have highly toxic effects on cells, increasing the apoptosis rate and reducing viability in the 3t3 and WiDr cell lines. Our in vivo analysis reveals that PS-NPs impair glucose-metabolism-related genes such as PI3K, AKT, PEPCK, PK, and GLUT2, leading to inflammation, oxidative stress, and liver injury. PS-NPs also increase several pro-apoptosis factiors, such as caspase-9, caspase-3, Bax, and Fas expression. The PS-NP treatment led to liver injury, as evidenced by the elevation of ALT and AST levels, edema, necrosis, and inflammatory cells. PS-NPs can be categorized as hazardous pollutants that affect human health, and we need to minimize daily plastic usage; this is particularly urgent due to the toxicity of these materials and their effects on glucose metabolism disorder, apoptosis, and liver injury. These results highlight that PS-NP exposure induces liver injury due to the dual impact of apoptosis and the disruption of glucose metabolism. This study emphasizes the need to minimize plastic consumption to ensure a good quality of life. Future studies should continue this research with the use of image diagnostics such as fluorescent microscopy and transmission electron microscopy (TEM) to determine the internalization of PS-NPs in liver cell lines.

## Figures and Tables

**Figure 1 ijms-26-04870-f001:**
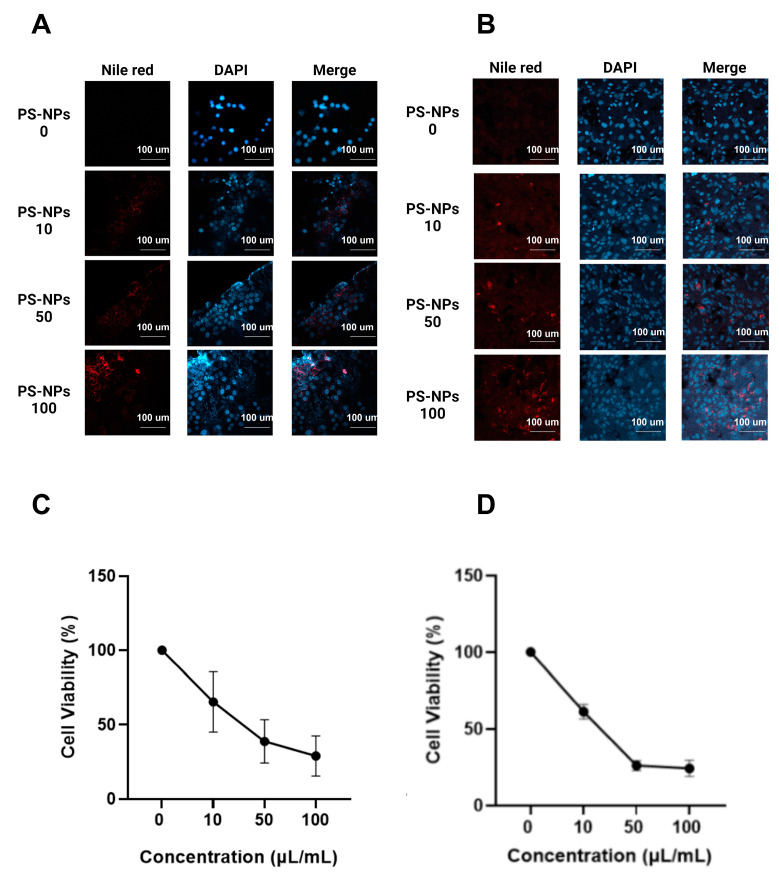
Effect of PS-NPs induction on 3t3 (**A**) and WiDr (**B**) cell lines, observed using a confocal microscope with Nile Red and DAPI staining. Percentage cell viability of 3t3 (**C**) and WiDr (**D**) cell lines after the MTT test. Nile red staining (red) indicates PS-NPs. DAPI staining (blue) displays the 3t3 and WiDr cell lines.

**Figure 2 ijms-26-04870-f002:**
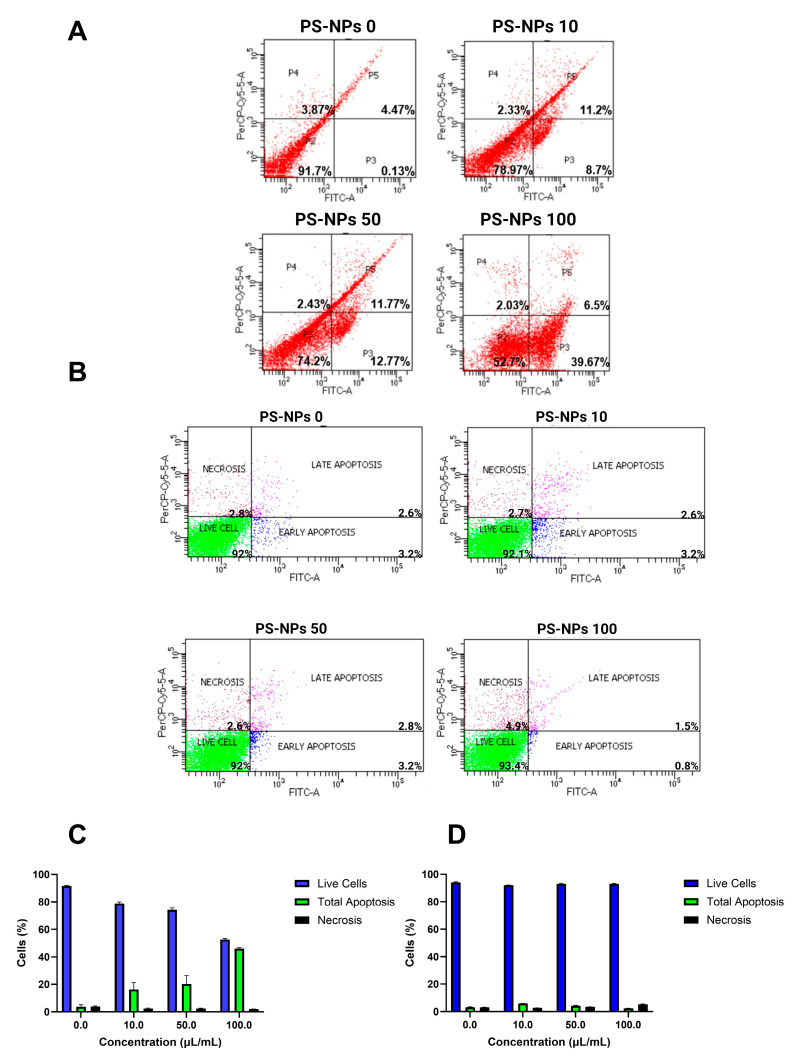
Flow cytometry test of PS-NPs induction on 3t3 (**A**) and WiDr (**B**) cell lines after staining with Annexin-PI. Percentage of total apoptotic (early apoptosis + late apoptosis) and necrotic cells from 3t3 (**C**) and WiDr (**D**) cell lines after induction by PS-NPs. Flow cytometry analysis: low left (viability percentage), low right (early apoptosis percentage), top right (necrosis percentage), and top left (late apoptosis percentage).

**Figure 3 ijms-26-04870-f003:**
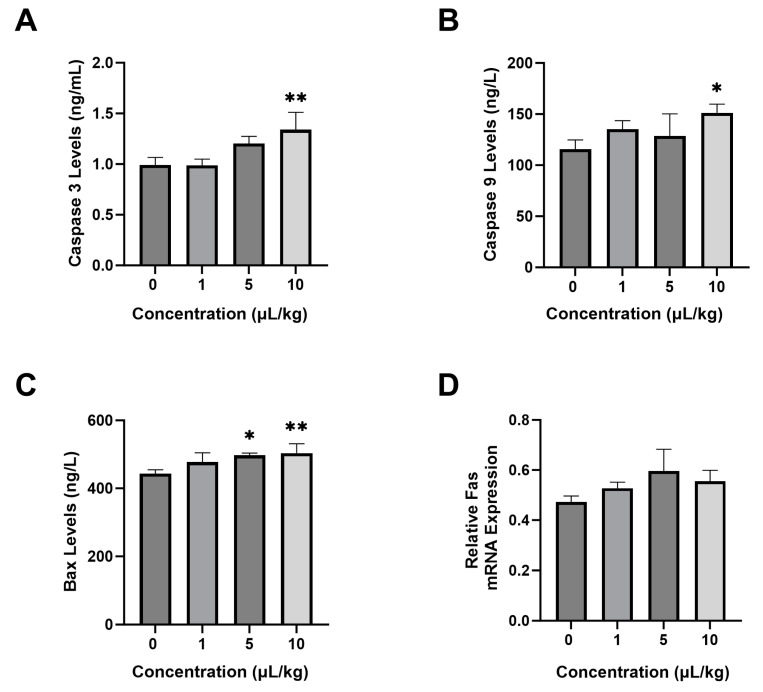
Effect of PS-NPs on pro-apoptosis parameters in rats. (**A**) Caspase-3 levels, (**B**) Caspase-9 levels, (**C**) Bax levels, and (**D**) Fas expression. Data are presented as the mean ± SD (*n* = 6 in each group). The bar displays the Tukey test analysis. Different superscripts show significant differences after analysis. * *p* < 0.05 compared with the K group. ** *p* < 0.01 compared with the control group.

**Figure 4 ijms-26-04870-f004:**
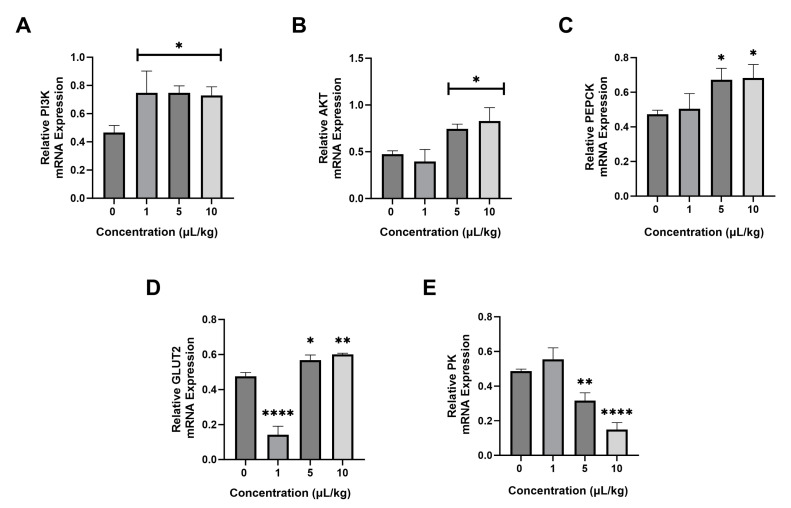
Effect of PS-NPs on the mRNA expression of glucose metabolism in rats. (**A**) PI3K expression, (**B**) Akt expression, (**C**) PEPCK expression, (**D**) GLUT2 expression, and (**E**) PK expression. Data are presented as the mean ± SD (n = 6 in each group). The bar displays the Tukey test analysis. Different superscripts show significant differences after analysis. * *p* < 0.05 compared with the control group. ** *p* < 0.01 compared with the control group. **** *p* < 0.0001. PI3K: Phosphoinositide-3-kinase. AKT: protein kinase B. PEPCK: phosphoenolpyruvate carboxykinase. GLUT2: glucose transporter-2. PK: pyruvate kinase.

**Figure 5 ijms-26-04870-f005:**
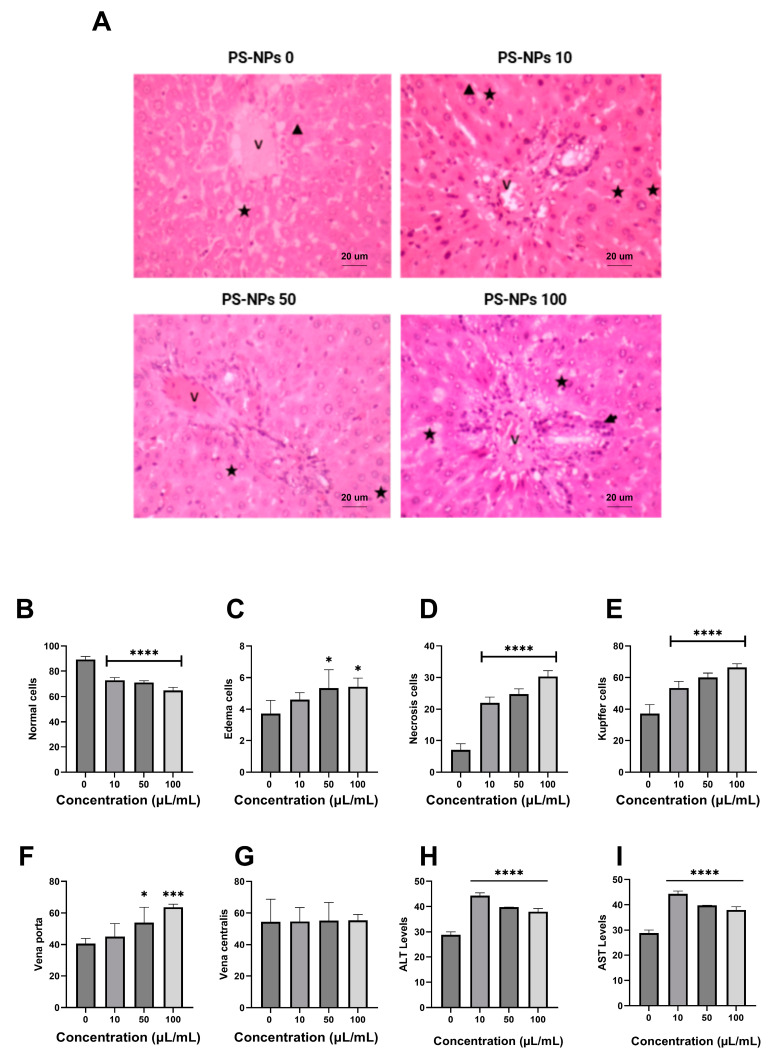
Effect of PS-NPs on liver injury in rats. (**A**) Histology of liver injury with HE staining at 200× magnification. The indication of liver injury was revealed in several indicator injuries, such as edema cells (triangle), necrosis cells (star), and inflammation cells (arrow). V: vena centralis. (**B**) normal cells, (**C**) edema cells, (**D**) necrosis cells, (**E**) Kupffer cells, (**F**) vena porta, and (**G**) vena centralis. Effect of PS-NPs on ALT levels (**H**) and AST levels (**I**). Data are presented as the mean ± SD (n = 6 in each group). The bar displays the Tukey test analysis. Different superscripts show significant differences after analysis. * *p* < 0.05 compared with the control group. *** *p* < 0.001 compared with the control group. **** *p* < 0.0001.

**Table 1 ijms-26-04870-t001:** List of primers for glucose metabolism and apoptosis parameters.

Gene	Sequence	
	Forward Primer	Reverse Primer
*GAPDH*	5′-TGC ACC ACC AAC TGC TTA GC-3′	5′-GGA TGC AGG GAT GAT GTT CT-3′
*AKT*	5′-GGA GCT CTG TTA GCA CCG TT-3′	5′-AGT GGA AAT CCA GTT CCG AGC-3′
*PI3K*	5′-ACA TCG ACC TAC ACT TGG GG-3′	5′-TCC CCT CTC CCC AGT AGT TT-3′
*FAS*	5′-CAG GAA CAA CTC ATC CGT TCT CT-3′	5′-GGA CCG AGT AAT GCC GTT CA-3′
*GLUT2*	5′-CCA GCA CAT ACG ACA CCA GAC G-3′	5′-CCA ACA TGG CTT TGA TCC TTC C-3′
*PK*	5′-CGT GGA CGA TGG GCT CAT CT-3′	5′-AGG TTC ACG CCC TTC TTG CT-3′
*PEPCK*	5′-GTC CCC CTT GTC TAC GAA GC-3′	5′-TGC ATG ATG ACC TTG CCC TTA-3′

## Data Availability

All of data produced in this study are available upon request from the corresponding authors.

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
