# Peer review of "Adverse Effect of Polystyrene Nanoplastics in Impairing Glucose Metabolism in Liver Injury"

_ijms, 2025, doi:10.3390/ijms26104870_

Round 1
Reviewer 1 Report
Comments and Suggestions for Authors
Title:Adverse effect of polystyrene nanoplastics in impairing glu-cose metabolism for liver injury
This study focuses on the effects of polystyrene nanoplastics (PS-NPs) on liver injury and glucose metabolism. This study comprehensively employs in vitro cell experiments (such as confocal microscopy, MTT assay, flow cytometry) and in vivo animal experiments (rat experiments), and combines multiple analytical techniques (ELISA, real-time PCR, histological analysis). The method is relatively comprehensive, and the influence of PS-NPs is deeply explored from multiple levels. This topic has high practical significance and scientific value. It is recommended to publish it after minor revisions.
- Although there have been certain introductions to plastic pollution and the hazards of PS-NPs, the specific distribution of PS-NPs in the environment, its sources, and the main ways it enters the human body can be further elaborated in detail to enhance the completeness and persuasiveness of the background.
- Figure 1. A-B,It is recommended to include a scale.
- Figure 2.A The flow cytometry cluster is inconsistent. It is recommended to check the original cell cluster.
- Figure 5. A,The HE diagram suggests adding a scale.
- In the Introduction and Discussion sections, for the introduction and discussion of Plastics and PS-NPs, the suggested references are: PMID:38064284 and PMID:36334886.
- Some of the results only present the data changes and lack in-depth explanations of the mechanisms behind the data. For example, when describing changes in gene expression, it should be further explained how these changes in gene expression specifically affect glucose metabolism and liver damage.
- Although there have been certain discussions on the mechanism by which PS-NPs affects apoptosis and glucose metabolism, it can be further explored in depth.
- For the reagents and instruments used, more detailed information should be provided, such as the purity of the reagents, the model and manufacturer of the instruments, etc., to enhance the repeatability of the experiment.
- The conclusion section can summarize the main findings of the research more concisely and clearly, highlighting the importance and significance of the research.
- Carefully check the grammar and spelling mistakes in the article to ensure the accuracy of the language expression.
Carefully check the grammar and spelling mistakes in the article to ensure the accuracy of the language expression.
Reviewer 2 Report
Comments and Suggestions for Authors
- The manuscript is poorly written. The Authors should consider correcting the English language throughout the manuscript.
- The authors provided limited information on the PS-NPs. Important characteristics such as particle size and surface charge should be discussed. Why were comparisons between particles of different sizes or surface charges (e.g., bare, negatively charged, and positively charged) not included?
- The nanoparticles were purchased; the authors should discuss how their properties and behavior relate to the effects observed concerning human exposure to relevant environmental or occupational particles.
- Authors should include high magnification images to show evident particle internalization into the cells.
- In Fig 1C, why were the standard deviations so high? This indicates too much variation between experiments.
- What is the relevance of choosing these two cell lines, 3T3 and WiDr, for the study? Why was a hepatic cell line not included?
- Authors overstated the results regarding apoptosis in WiDr cells, where there was no/minimal apoptotic induction compared to the control. What is the viability mentioned in Fig 2C and 2D?
- Was there any evidence of particle internalization in liver tissue? If there is, the authors should present this data to strengthen the connection between particle exposure and the observed effects.
- Why was the MTT absorbance measured at 450 nm instead of 590 nm?
- In the Flow cytometry apoptosis assay, it was mentioned that cells were suspended in PI buffer containing PI, and again, the sample was added with annexin-PI. Authors should check this.
The manuscript is poorly written. The Authors should consider correcting the English language throughout the manuscript.
